# Fluxes, patterns and sources of phosphorus deposition in an urban-rural transition region in Southwest China

Yuanyuan Chen[1], Jiang Liu[2], Jiangyou Ran[1], Rong Huang[1], Xuesong Gao[1], Wei Zhou[1], Ting Lan[1], Dinghua Ou[1], Yan He[3], Yalan Xiong[1], Ling Luo[3], Lu Wang[4], Ouping Deng[1,*]

[1] College of Resources, Sichuan Agricultural University, Chengdu 611130, P.R. China

[2] State Key Laboratory of Environmental Geochemistry, Institute of Geochemistry, Chinese Academy of Sciences, Guiyang, 550081, P.R. China

[3] College of Environmental Sciences, Sichuan Agricultural University, Chengdu 611130, P.R. China

[4] Chongzhou Meteorological Bureau, Chengdu 611230, P.R. China

**Correspondence:** Ouping Deng (ouping@sicau.edu.cn)

**Abstract:** Understanding the patterns of atmospheric phosphorus (P) deposition is essential for assessing the global P biogeochemical cycle. The atmospheric phosphorus (P) is an essential source of P in agricultural activities as well as eutrophication in waters; however, the information on P deposition is relatively less paid attention, especially in the anthropogenic influencing region. Therefore, this study chose a typical urban-rural transition as a representative to monitor the dry and wet P depositions for two years. The results showed that the fluxes of atmospheric total P deposition ranged from 0.50 to 1.06 kg P $hm^{-2}$ $yr^{-1}$, and the primary form was atmospheric dry P deposition (76.13%, 0.76~0.84 kg $hm^{-2}$ $yr^{-1}$). Moreover, it was found that the monthly variations of P deposition were strongly influenced by meteorological factors, including precipitation, temperature, and relative humidity. However, the fluxes of dry P deposition and total P deposition were more affected by land use, which increased with the agro-facility, town, and paddy field areas, but decreased with the forest and country road areas. These findings suggested that dry P deposition was the primary form of total P deposition, and P deposition could be both affected by meteorological factors and land use types. Thus, proper management of land use may help mitigate the pollution caused by P deposition.

## 1 Introduction

Phosphorus (P) is generally considered the essential nutrient and growth-limiting element in terrestrial and aquatic ecosystems (Vitousek et al., 2010; Peñuelas et al., 2013). Over the past few decades, with the increasing application of P fertilizers and fossil fuel

combustion, substantial anthropogenic P has been emitted into the atmosphere (Wang et al., 2015; Du et al., 2016). Moreover, the deposition of atmospheric P on terrestrial surfaces overfertilizes some natural and seminatural ecosystems (Camarero and Catalan, 2012; Cleveland et al., 2013; Wang et al., 2015), especially aquatic ecosystems (Pollman et al., 2002; Tong et al., 2017). However, P deficiency was also found in a large proportion (43%) of land area, in which P-input, such as deposition, will significantly increase the productivity of plants (Elser et al., 2007; Du et al., 2020; Hou et al., 2020). Hence, estimating the deposition characteristics of atmospheric P is important to understand the biogeochemical process of P and could provide information on water nutrient pollution control.

Several research efforts have quantified P deposition fluxes from the field scale to the global scale, and the results showed large uncertainty. For instance, a recent meta-analysis of 394 sites from a global scale covering the period 1959–2020 observed that the average value of atmospheric total P deposition was 0.58±0.72 kg $hm^{-2}$ $yr^{-1}$ (Pan et al., 2021). It has been reported that total P deposition fluxes range from 0.002 to 2.53 kg $hm^{-2}$ $yr^{-1}$ at 41 in-situ sites across China (Zhu et al., 2016). In addition, the overall average fluxes of total P deposition during 2008–2018 at four sites located in Southwest China ranged from 0.12 to 4.15 kg $hm^{-2}$ $yr^{-1}$ (Song et al., 2022). Previous studies have identified that P deposition rates vary at local scales. Therefore, P depositions exist in temporal and spatial variations at a regional scale, and measurements across different areas are needed to better understand the role of P deposition in the global P cycle.

Different land-use types and so caused landscape perturbations largely determine P deposition (Peñuelas et al., 2011). For instance, the application of P fertilizer can be the main source of higher P deposition in agricultural areas (Winter et al., 2002; Anderson and Downing, 2006). At rural sites, biogenic sources are the primary contributor to atmospheric P deposition, whereas anthropogenic sources (such as the application of P fertilizer) have a larger effect on atmospheric P deposition at suburban sites in Japan (Chiwa, 2020). Additionally, a previous study revealed that sites characterized by land-use types, such as agro-facility contributed more P deposition (3.22 kg $hm^{-2}$ $yr^{-1}$), which was higher than in rural, urban, and forest areas (0.20 ~1.07 kg $hm^{-2}$ $yr^{-1}$, Song et al., 2022). Besides, P depositions in forested sites significantly increase with decreasing distance to the nearest large cities (Du et al., 2016). Additionally, field studies have observed that the seasonal distribution pattern of precipitation is a factor influencing temporal variations of atmospheric P deposition (Tsukuda et al., 2005; Zhu et al., 2016; Chiwa et al., 2020). However, P deposition had no relationship to precipitation, but a significantly positive dependence on temperature was observed (Tipping et al., 2014). There is still great uncertainty about how these influencing factors affect the variation in P deposition. Further comprehensive identification of the variation drivers of atmospheric P deposition on a regional scale is needed.

Atmospheric P mainly occurs in the form of aerosols rather than in a stable gaseous phase (Mahowald et al., 2008). Hence, larger and heavier P-containing aerosols are mainly contributed by local sources because they can only be transported over short distances, while

fine dust can originate from thousands of kilometers (Tipping et al., 2014). Besides, different P-containing aerosols are likely to deposit to the terrestrial surface in distinct ways. Atmospheric P-containing aerosols that were scavenged in and below clouds by precipitation and deposited on the terrestrial surface were defined as wet deposition (Yang et al., 2012). These were removed and deposited onto the terrestrial surface by the adsorption of water droplets under the action of gravity, which was defined as dry deposition (Grantz et al., 2003). However, most reported measurements are based on bulk deposition, which includes wet deposition plus a fraction of dry deposition. Meanwhile, the measurements of dry deposition are quite sparse. Hence, it's essential to collect wet deposition and dry deposition separately, which can enrich the P database and clarify the global P deposition pattern.

Urban-rural transition regions were formed commonly in the process of urbanization and were deeply influenced by human beings. The patterns and sources of P deposition are more complex here than in natural ecosystems. However, in-situ P deposition studies are limited here. Therefore, a typical urban-rural transition region in southwestern China was selected, and 2-years monitoring of wet and dry P depositions in this region was conducted. The aims of this study are (1) to determine the spatial and temporal characteristics of P deposition fluxes in urban-rural transition areas; (2) to identify the factors affecting P deposition fluxes in urban-rural transition areas; and (3) to reveal the "source/sink" relationship between P deposition and local land use. The results of this study are important for understanding the process of regional P deposition and regional P management with "source/sink" land use.

## 2 Materials and methods

### 2.1 Sampling sites

This study was conducted from March 2015 to February 2017 at nine sites that were chosen to explore atmospheric P deposition spanning a transect covering urban areas (UA), intensive agricultural areas (IAA), and rural areas (RA) in the southwestern Chengdu Plain (Fig. 1, Table S1, Deng et al., 2019). Urban areas, including Shangnan Street (SS), Yongkang Street (YS), and Xihe Bridge (XB) sites, are located in Chongzhou, which has 74.4 $km^2$ of urban land and 130,000 permanent dwellers. Intensive agricultural areas, including the Caichang (CC), Liaoyuan (LY), and Qiquan (QQ) sites, covered 1.8 $km^2$ of the agro-facility land-use type, which accounted for approximately 69.9% of the total in nine sites. Rural areas, including Yuantong (YT), Liujie (LJ), and Huaiyuan (HY) sites, covered 13.59 $km^2$ of forest, which accounted for approximately 96.2% of the total in nine sites. More details about the study sites are shown in Table 1. The climate at the sites is subtropical monsoon humid, with monthly precipitation, ambient temperature, relative humidity, and wind speed at 9 sites varying from 0.6 to 238.63 mm, 5.83 to 27.3°C, 66.0% to 89.3%, and 0.5 to 1.80 m $s^{-1}$, respectively. The meteorological data used in this study are from the Chongzhou Meteorological Bureau, Sichuan Province, China.

### 2.2 Sample collection and analysis

Both dry deposition (from gases, aerosols, and particles) and wet deposition (from rain and snow) of P were monitored. Dry deposition was determined by the aqueous surface

method (Anderson and Downing, 2006). Briefly, three pre-clean glass cylinders (inner diameter × height of 10.5 cm × 14.5 cm) were used as dry collectors at each site. All the collectors were placed 1.2 m above the ground with no obstacles and tall buildings around each site. A stainless-steel net (pore size, 0.02 × 0.02 $m^2$) was used to avoid any disturbance and pollution from birds and crops. Ultra-pure water was filled into each collector, and the water depth was kept at approximately 10 cm (Wang et al., 2016). Five consecutive non-rainy days at the end of each month were used to collect samples. A cover on the top of the collectors was manually closed during rainfall events to eliminate influences from wet deposition. Water samples mixed with dry depositions were transferred into pre-clean glass bottles and transported to the laboratory to determine total P (TP) concentrations on the same day.

In addition to dry deposition, three parallel collectors were used at each site to collect wet deposition. Wet deposition was only collected during rainfall events (Oladosu et al., 2017). If the volume of samples (100 mL) collected in one rainfall event was not enough for in-lab measurements, samples from several rainfall events were pooled as one mixed sample. The duration (min) and rainfall capacity (mm) were recorded for each rainfall event. Rainfall samples were transferred to glass bottles with lids (100 mL) and stored in a cooler. Changes in sample volume and air exposure were minimized. Moreover, river water samples from the Xihe River at site XB were collected to measure the P concentration.

The total P in the collected samples was digested using potassium persulfate at 120 °C to convert TP to $PO_4^{3-}$ and then analyzed $PO_4^{3-}$ using ammonium molybdate by using an ultraviolet spectrophotometer at 700 nm.

### 2.3 Calculations of MDP, MWP, and MTP

Monthly dry deposition (MDP) was calculated as the product of the amount of sampling fluid and the concentrations of TP in the sampling fluid.

$$\text{MDP (kg P hm}^{-2}) = \frac{C_d \times V_d \times N}{S \times 10^5 \times 5} \quad (1)$$

where $C_d$ is the concentration of TP in the monthly sampling fluid, mg P $L^{-1}$; $V_d$ is the sampling fluid amount, mL; d represents each month; N is the number of non-rainy days per month, d; S is the surface area of the sampling cylinder, $m^2$; and 5 is the sampling days per month.

Monthly wet deposition (MWP) was calculated as the product of the monthly precipitation amount and the concentrations of P types in wet precipitation.

$$\text{MWP (kg P hm}^{-2}) = 0.01 \times C_i \times P_i \quad (2)$$

where $C_i$ is the concentration of TP in monthly wet precipitation, which was mixed with all samples for a month, mg P $L^{-1}$; $P_i$ is the monthly precipitation amount, mm; and i represents each month.

Monthly total deposition (MTP) is the sum of MDP and MWP.

$$\text{MTP (kg P hm}^{-2}) = \text{MDP} + \text{MWP} \quad (3)$$

where MDP and MWP are calculated from (1) and (2).

### 2.4 Land use data and analysis

The land use data (2016) used in this study were provided by the Center of Land Acquisition and Consolidation in Sichuan Province. Land-use types were divided as follows: agricultural area (paddy field, dry farm, yard, and agro-facility area), build-up area (urban, town, and village), and road (highway and country road), forest, and water (Fig. 1). Taking the sampling point as the center and extracting the land use type area with a radius of 5 kilometers from the center, ArcGIS 10.6 was used. Correlation analysis was used to study the covariation between the fluxes of atmospheric total, dry, and wet P deposition, and land use areas.

### 2.5 Statistical analyses

To control the quality of field monitoring experiments, three replicates were set up at each point. One-way analysis of variance (ANOVA) was performed to determine the spatial and temporal variation among the three areas. Statistically significant differences were set at $P < 0.05$. Pearson correlation analysis with a two-tailed significance test was used to examine the relationship between the fluxes of atmospheric total, dry, and wet P deposition, and land-use types, and meteorological factors. All analyses were conducted using SPSS® 20.0 (SPSS Inc., Chicago, USA).

## 3 Results

### 3.1 Monthly variations of P deposition and its constituents

Monthly variations of atmospheric total, dry, and wet P deposition fluxes at nine study

sites were monitored for 24 months (Fig. 2). For wet deposition, the fluxes peaked in July 2016 (0.06~0.15 kg P $hm^{-2}$ $mon^{-1}$), and the lowest fluxes were found in February 2015 (0.00~0.00 kg P $hm^{-2}$ $mon^{-1}$) (Fig. 2a). In contrast, the highest fluxes of dry P deposition occurred in November 2015 (0.07~0.24 kg P $hm^{-2}$ $mon^{-1}$), and the lowest values occurred in April 2016 (0.01~0.02 kg P $hm^{-2}$ $mon^{-1}$) (Fig. 2b). A similar variation trend was observed in total P deposition, but it reached its lowest value in April 2015 (0.01~0.03 kg P $hm^{-2}$ $mon^{-1}$) (Fig. 2c). Additionally, the monthly contribution rates of dry P deposition to total P deposition varied from 24.98% to 99.72% temporally (Fig. 3). Atmospheric dry P deposition generally constituted more than half of the total P deposition, except in April and August 2015 and May and July 2016, in which heavy rains accounting for 20.37% of the total precipitation were observed.

**3.2 Seasonal variations of P deposition**

The fluxes of atmospheric wet P deposition in summer are 2.5~17.1 times higher than those in other seasons ($P < 0.05$, Fig. 4a). Conversely, the fluxes of dry P deposition and total P deposition in autumn are significantly higher than those in other seasons (by 1.4~2.9 times, $P < 0.05$, Figs. 4b and 4c). Summer (June to August) is the key season for wet P deposition, while autumn (September to November) is the key season for dry P deposition and total P deposition. The study area belongs to the subtropical monsoon climate zone, with high rainfall, temperature, and humidity in summer and autumn, which contribute to the emission and deposition of P. Thus, correlation analysis between three types of depositions and meteorological factors (precipitation, wind speed, temperature, and relative humidity) was

adopted. The fluxes of wet P deposition were positively correlated with precipitation ($R=0.917$) and temperature ($R=574$) ($P < 0.01$). While the values of dry P deposition have a positive correlation with relative humidity ($R=0.439$) ($P < 0.01$). Additionally, significant correlations were found between total P deposition fluxes and precipitation ($R=0.360$), relative humidity ($R=0.481$) ($P < 0.01$), and temperature ($R=0.294$) ($P < 0.05$) (Table S2).

**3.3 Spatial variation of annual P deposition among nine sites**

The average atmospheric wet P deposition rates at the nine sites showed no significant spatial variations (Fig. 5a), whereas the dry P deposition and total P deposition were observed to have significant spatial variations across the study urban-rural transition (Fig. 5b, c) ($P < 0.05$). Specifically, the annual fluxes of dry P deposition in CC, LY, and QQ (0.76~0.84 kg $hm^{-2}$ $yr^{-1}$) were significantly higher than those in SS, YS, and XB, and YT, LJ, and HY (0.32~0.49 kg $hm^{-2}$ $yr^{-1}$) ($P < 0.05$). The average rate of dry P deposition among the nine sites was 0.54±0.18 kg P $hm^{-2}$ $yr^{-1}$. In addition, the annual fluxes of total P deposition decreased in the order CC, LY and QQ (0.97~1.06 kg $hm^{-2}$ $yr^{-1}$) > SS, YS and XB (0.61~0.71 kg $hm^{-2}$ $yr^{-1}$) > YT, LJ and HY (0.50~0.55 kg $hm^{-2}$ $yr^{-1}$).

**3.4 Relationship between land use types and P deposition.**

To better understand the potential sources of P deposition, the correlations between monthly fluxes of P deposition and areas of land-use types were analyzed (Fig. 6). The monthly atmospheric wet P deposition fluxes were significantly positively correlated with the agro-facility areas in the five months ($P < 0.05$) (Fig. 6a), and the monthly dry P deposition

and total P deposition fluxes were significantly positively correlated with the agro-facility areas for almost the whole year ($P < 0.05$) (Fig. 6b, c). Meanwhile, the annual fluxes of atmospheric total, dry, and wet P deposition all were strongly positively correlated with the agro-facility areas (R=0.765, R=0.898, and R=0.903, $P < 0.05$).

In addition, dry P deposition and total P deposition both had a positive correlation with the town and paddy field during almost the whole year; the town was significant in February and October, and another was significant in September ($P < 0.05$). Conversely, there was a negative correlation with country roads and forests during the whole year, with country roads being significant in November ($P < 0.05$).

# 4 Discussion

## 4.1 Temporal variability of P deposition

More atmospheric P was deposited through wet deposition in summer than in other seasons ($P < 0.05$). This phenomenon occurred due to the fluxes of atmospheric wet P deposition having a significantly positive correlation with monthly precipitation and temperature (Table. S2). In this study, area, summer accounted for approximately 51.46% of the annual precipitation, which would allow more P-containing aerosols to be scavenged in and below clouds by precipitation and deposited on the terrestrial surface, resulting in higher fluxes in summer as well. Similarly, precipitation did have a positive impact on the monthly P deposition fluxes in previous studies (Tsukuda et al., 2005; Zhu et al., 2016; Wang et al., 2018). In addition, the temperature in summer was approximately 1.42~3.11 times higher

than that in other seasons. High temperatures can decrease the stability of the atmosphere and increase the activity of P-containing aerosols, which can enlarge their contact area with the atmosphere. This causes more aerosols containing P to be adsorbed and dissolved in the air. (Huang et al., 2011).

The fluxes of dry P deposition showed varied seasonal variation with those of wet P deposition and had the highest values in autumn than in other seasons ($P < 0.05$) (Fig. 5). Previous studies found that P-containing aerosols were the main components of dry P deposition collected by the alternative surface method and were changed with relative humidity (Qi et al., 2005). In this study, the fluxes of dry P deposition were strongly correlated with relative humidity (Table. S2). An increase in relative humidity would lead to an enlargement in particle size and an increase in hygroscopic growth. This growth can significantly increase the particle deposition rate (Mohan, 2016). There are two reasons as follows: on the one hand, P-containing aerosol contact areas with water droplets will be enlarged, which will cause more aerosols to deposit. On the other hand, aerosols can absorb more moisture and increase particle size (hygroscopic growth), making them deposit quickly. Overall, wet deposition was affected by precipitation and temperature. In contrast, relative humidity was the main driving force for dry deposition.

## 4.2 Analysis of deposition composition characteristics

Several studies divided P deposition into dry and wet deposition separately for monitoring, and the results demonstrated that the percentage of deposition was in the range of 50~85% (Hou et al., 2012; Tipping et al., 2014). Similarly, this phenomenon was observed in

this study. To explain this, first, only a fraction of P-containing aerosols was water-soluble (Herut et al., 2005; Nenes et al., 2011), causing it to likely be deposited as dry deposition. Second, it was observed that the months dominated by wet deposition all followed higher precipitation during the whole year. As discussed earlier, precipitation accelerates wet deposition. Third, the composition characteristics indicated various P sources. The fine dust from desert and soil is more likely to be transported over a long distance and deposited as wet deposition. However, it originated from intensively farmed, especially arable, soil fertilized with P and was more likely to be deposited as dry deposits. The factor is that the fraction of soil lost as dust is small and likely to be enriched, thus increasing the content of P-aerosols and increasing their size (Field et al., 2010). In general, the construction of depositions was impacted by the solubility of P depositions and meteorological factors.

### 4.3 Spatial variation of annual P deposition fluxes

Due to the varied mechanisms of wet and dry deposition processes, the fluxes of atmospheric wet and dry P deposition showed distinct spatial variation trends (Fig. 5). The annual fluxes of atmospheric P dry deposition in CC, LY, and QQ (0.76~0.84 kg $hm^{-2}$ $yr^{-1}$) were significantly higher than those in other areas (0.32~0.49 kg $hm^{-2}$ $yr^{-1}$) ($P < 0.05$), but the fluxes of wet P deposition did not show significant spatial variation.

In general, dry P deposition is dominated by local sources, while a considerable proportion of wet P deposition comes from long-distance P-particle sources (Mahowald et al., 2008; Das et al., 2011; Gross et al., 2016). In this study, more local P-aerosols were emitted into the atmosphere in CC, LY, and QQ for high-intensity agricultural production, such as

large-scale livestock and poultry breeding. These local P-aerosols were deposited as local resources, causing a higher value of dry P deposition. Hence, to further clarify the influencing factors from multiple land-use types on P deposition, the analysis of the correlation between land-use types and P deposition was carried out as follows in this study.

Moreover, the annual total P deposition fluxes in CC, LY, and QQ (0.97 ~1.06 kg $hm^{-2}$ $yr^{-1}$) were significantly higher than those in YT, LJ, and HY (0.50 ~0.55 kg $hm^{-2}$ $yr^{-1}$) (Fig. 4c; $P < 0.05$). It was also higher than a large number of fluxes on a global scale, such as in Chinese forests (0.69 kg P $hm^{-2}$ $yr^{-1}$, Du et al., 2016) and a French tropical forest (0.62 kg P $hm^{-2}$ $yr^{-1}$, Van Langenhove et al., 2020). Additionally, here, compared with the global total P deposition rates compiled from 396 published observations during the period 1959 to 2020, including in North America (0.26 kg $hm^{-2}$ $yr^{-1}$), Europe (0.29 kg $hm^{-2}$ $yr^{-1}$), Asia (0.41 kg $hm^{-2}$ $yr^{-1}$), Oceania (0.19 kg $hm^{-2}$ $yr^{-1}$), South America (0.40 kg P $hm^{-2}$ $yr^{-1}$), and Africa (0.58 kg P $hm^{-2}$ $yr^{-1}$) (Pan et al., 2021), the fluxes in CC, LY, and QQ in this study showed much higher values. To explain this phenomenon, first, the level of economic development and natural environment conditions in different regions varied from the region. For instance, in developed regions, substantial anthropogenic P has been emitted into the atmosphere and transported to surrounding areas with the application of P fertilizer on farmlands and the combustion of fuels (Wang et al., 2015; Du et al., 2016). Additionally, this study was compared with the prior findings that were all carried out under multiple land uses. On the one hand, it can also be noted that nearly all measurements above refer to natural or seminatural locations. On the other hand, the land-use types at the nine sites in this study

were different from each other. A previous study revealed that the sites characterized by land use in an agro-facility contributed more P deposition (3.22 kg $hm^{-2}$ $yr^{-1}$), which was higher than that in rural, urban, and forest areas (0.20 ~1.07 kg $hm^{-2}$ $yr^{-1}$, Ling, et al., 2022). This discrepancy could be explained by the collection methods used for P deposition. In this study, wet deposition and dry deposition samples were collected separately, while most reported measurements were based on bulk deposition, which generally ignored dry deposition (Helliwell et al., 1998; Tipping et al., 2014). Additionally, the actual rates could be underestimated with a decrease in collection frequency due to the evaporation of water that would remove the P deposition, which would also be on the wall of the cup. Therefore, differences in the method of sample collection could cause variability among the regions discussed above.

In general, several factors could contribute to the spatial variation of P deposition. Moreover, the potential risk of P deposition in this study area cannot be ignored. More attention needs to be paid to effectively managing P inputs and cycles.

### 4.4 Relationship between land use and phosphorus deposition

There are positive and negative correlations between fluxes of P deposition and land-use types. On the one hand, it was positively correlated with agro-facility areas, towns, and highways, which suggested that those land-use types might be a “source” for P deposition. On the other hand, there was a negative correlation with the country road and forest, which revealed that “sink” land use types for P deposition could be those.

This study showed that P deposition had a strong positive correlation with the agro-facility (Fig. 6). In this study, CC, LY, and QQ had approximately 4.5 times larger agro-facility areas than the other sites (Table 3.). Commonly, agro-facility areas include land designated for livestock and poultry breeding, fertilizer plants, greenhouses with vegetable production, and aquaculture (《Current land use classification》GB/T 21010-2007). The flux of total P deposition could be variable, and it mainly depended on local sources (Song et al., 2022). Accordingly, this study investigated local land-use types and found that the lack of light resources in the Chengdu Plain led to fewer greenhouses in the agro-facility areas and more livestock and poultry breeding and fertilizer industries. Livestock production and manure generation could be contributors to P deposition (Ma et al., 2011; Tong et al.,2017; Zhang et al., 2019). Many previous studies reported the sources of P deposition. For instance, P deposition originated from intensive agricultural management and extraction of rock phosphate in Sichuan suburban areas (Song et al., 2022). In areas with a high density of livestock husbandry in Germany, P deposition originated from agricultural emissions from livestock farming (Tipping et al., 2014). Furthermore, emissions from the phosphate fertilizer industry will cause high phosphorus concentrations in the air layer and increase total P deposition fluxes (Rodríguez et al., 2011). This study demonstrated that the land use of agro-facility acts as a main "source" affecting P deposition almost year-round.

In addition, the monthly fluxes of atmospheric dry P deposition and significantly positive correlation with the town area in February and October (Fig. 6b, c; $P < 0.05$) during the Spring Festival and National Day. In these important festivals, custom fireworks could

induce more harmful gases and dust, thereby increasing combustion emissions of dissolved P, dust emissions, and organic P contained in bioaerosol emissions entering the atmosphere (Kanakidou et al., 2020). In addition, monthly dry P deposition and total P deposition fluxes had a positive correlation with the paddy field during almost the whole year and were significant in the fertilizer period (September, $P < 0.05$). This phenomenon is mainly caused by agricultural phosphorus emissions to a certain extent. Agricultural activities have intensely disturbed paddy field disturbances (Anderson and Downing, 2006). P deposition may originate from local agricultural sources (Tipping et al., 2014). A previous study reported that P deposition increased after P fertilizer application (Gao et al., 2009).

Notably, monthly fluxes of dry P deposition and total P deposition both had a negative correlation with forest and country roads (Fig. 6b, c). It is well known that forests can absorb harmful gas, aerosols, and dust particles, including P-containing aerosols, which is attributed to the porous sponge-like underlying surface, high productivity, and strong microbial activity (Oladosu et al., 2017; Wang et al., 2017; Zhai et al., 2019). Likewise, forest canopies could decrease phosphorus deposition by trapping dust and particulates (Zhou et al., 2018). Due to similar reasons, paved country roads without hardening showed a similar correlation with P deposition.

In general, land use plays a vital role in P deposition. It was suggested that agro-facility, town, and paddy fields were "source" land types, while forest and country roads were "sink" for P deposition. Furthermore, the key land use for P deposition is the agro-facility in a typical urban-rural transition.

### 4.5 Management practice of regional P

According to a survey of surface water quality, the concentration of total P based on VWM in the Xihe River (0.10 ~0.23 mg $L^{-1}$) was higher than Grade III (0.2 mg $L^{-1}$) of the National Surface Water Quality Standard in China (GB3838-2002). Referring to the Standard, Grand IV water could only be used for industrial production or human amusement without direct body contact, and Category V water could only be used for agriculture or scenery (Tong et al., 2016). The upper reach of the Xihe River is a backup drinking water conservation area and needs to control P pollution and decrease P loads.

A previous study found that P deposition increasingly contributed to inland freshwater, e.g., rivers and lakes, and became the key process of regional P pollution management (Tong et al., 2017). Given the main results of this study, adjusting the land use structure is the first step for P management. Increasing areas of forests and controlling the scale of aquaculture and livestock farming. The next step is to increase the use of ecological materials and reduce road hardening in the process of road construction. Third, to manage fertilization effectively, more attention should be given to four major fertilization factors (the 4Rs): right rate, right source, right placement, and right timing. Last, a policy of prohibition and restriction on fireworks should be implemented. Therefore, more flexible regional strategies need to be applied to address the different temporal and spatial trends and sources of P deposition.

## 5 Conclusion

From the perspective of temporal and spatial analysis, an in-situ study was carried out to understand the patterns of atmospheric P deposition in this region. The first major finding is that P deposition showed seasonal variability under the influence of meteorological factors. Wet deposition is impacted by precipitation and temperature, making its fluxes significantly higher in summer than in other seasons ($P < 0.05$). While dry deposition is affected by relative humidity, it was significantly higher in autumn ($P < 0.05$). The results showed that dry deposition is the main contributor to total deposition. Furthermore, the monthly fluxes of dry P deposition present a significant spatial variation under different land-use types ($P < 0.05$) because UAA, including the CC, LY, and QQ sites, could emit more particulate P depositions, which would result in more dry depositions. Based on correlation analysis, it was found that "source" land use might be agro-facility, town, and paddy field areas, while "sink" land use might be forest and country road areas. Thus, to effectively control regional P, the "source/sink" relationship between P deposition and land-use types should be considered.

**Author contribution**

Conceptualization, Methodology, and Writing original draft (YYC, JL, OPD), Visualization and Validation (JYR, RH), Review & editing (RH, XSG, WZ), Methodology (DHO, TL), Data curation (YH, YLX, LW, LL), Funding acquisition, Writing - review & editing (OPD).

**Competing interests**

The authors declare that they have no known competing financial interests or personal

relationships that could have appeared to influence the work reported in this paper.

**Acknowledgments**

This study was supported by the Department of Science and Technology of Sichuan Province, China [2020YFH0163, 2021YFS0277], and the National Natural Science Foundation of China [42007212, 42107247].

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

## Tables

**Table 1** The types of land use and those areas were divided as follows: agricultural area (paddy field, dry land, yard, and agro-facility area), build-up area (urban, town, and village), road (highway and country road), water, and forest.

| Classification | Site code | Paddy field | Dry land | Yard | Agro-facility area | Urban | Town | Village | Highway | Country road | Water | Forest |
|---|---|---|---|---|---|---|---|---|---|---|---|---|
| UA | SS | 31.12 | 3.28 | 0 | 0.08 | 25.13 | 1.04 | 8.92 | 2.57 | 0.27 | 5.15 | 0.05 |
| | YS | 27.71 | 2.58 | 0 | 0.08 | 27.54 | 2.79 | 7.75 | 3.43 | 0.23 | 5.34 | 0.15 |
| | XB | 34.20 | 2.74 | 0 | 0.22 | 21.68 | 1.51 | 8.65 | 2.62 | 0.23 | 5.55 | 0.02 |
| IAA | LY | 52.49 | 4.25 | 0.06 | 0.49 | 0 | 5.42 | 11.44 | 1.36 | 0.16 | 1.3 | 0.07 |
| | QQ | 55.00 | 4.13 | 0.49 | 0.72 | 0 | 1.75 | 12.04 | 0.96 | 0.03 | 2.85 | 0.07 |
| | CC | 41.43 | 5.98 | 0.16 | 0.6 | 0 | 8.00 | 11.60 | 2.00 | 0.04 | 6.86 | 0.18 |
| RA | LJ | 41.04 | 20.51 | 0.11 | 0.2 | 0 | 2.16 | 11.90 | 0.22 | 0.52 | 1.26 | 0.13 |
| | YT | 51.14 | 7.01 | 0 | 0.09 | 0 | 2.42 | 11.46 | 0.84 | 0.01 | 3.86 | 0.33 |
| | HY | 38.88 | 5.61 | 1.57 | 0.11 | 0 | 3.20 | 10.46 | 1.20 | 0.21 | 2.75 | 13.13 |

[a]SS, Shangnan Street; YS, Yuantong Street; XB, Xihe Bridge; CC, Caichang; LY, Liaoyuan; QQ, Qiquan; YT, Yuantong; LJ, Liujie; HY, Huaiyuan. [b]IAA, intensive agricultural area; RA, rural area.

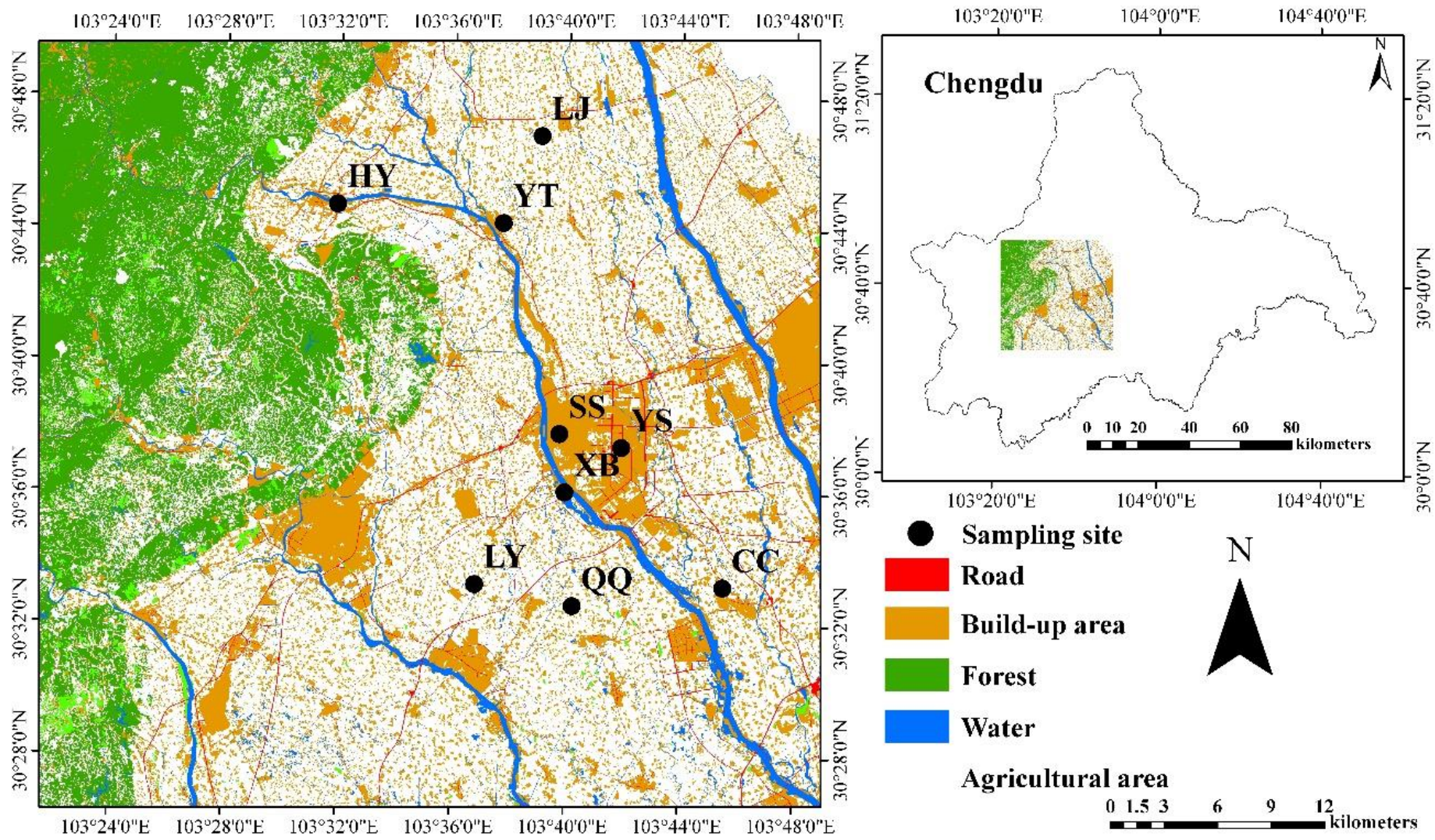

**Figure 1:** Location of the sampling sites. SS, Shangnan Street; YS, Yongkang Street; XB, Xihe Bridge; CC, Caichang; LY, Liaoyuan; QQ, Qiquan; YT, Yuantong; LJ, Liujie; HY, Huaiyuan. (Deng et al., 2019)

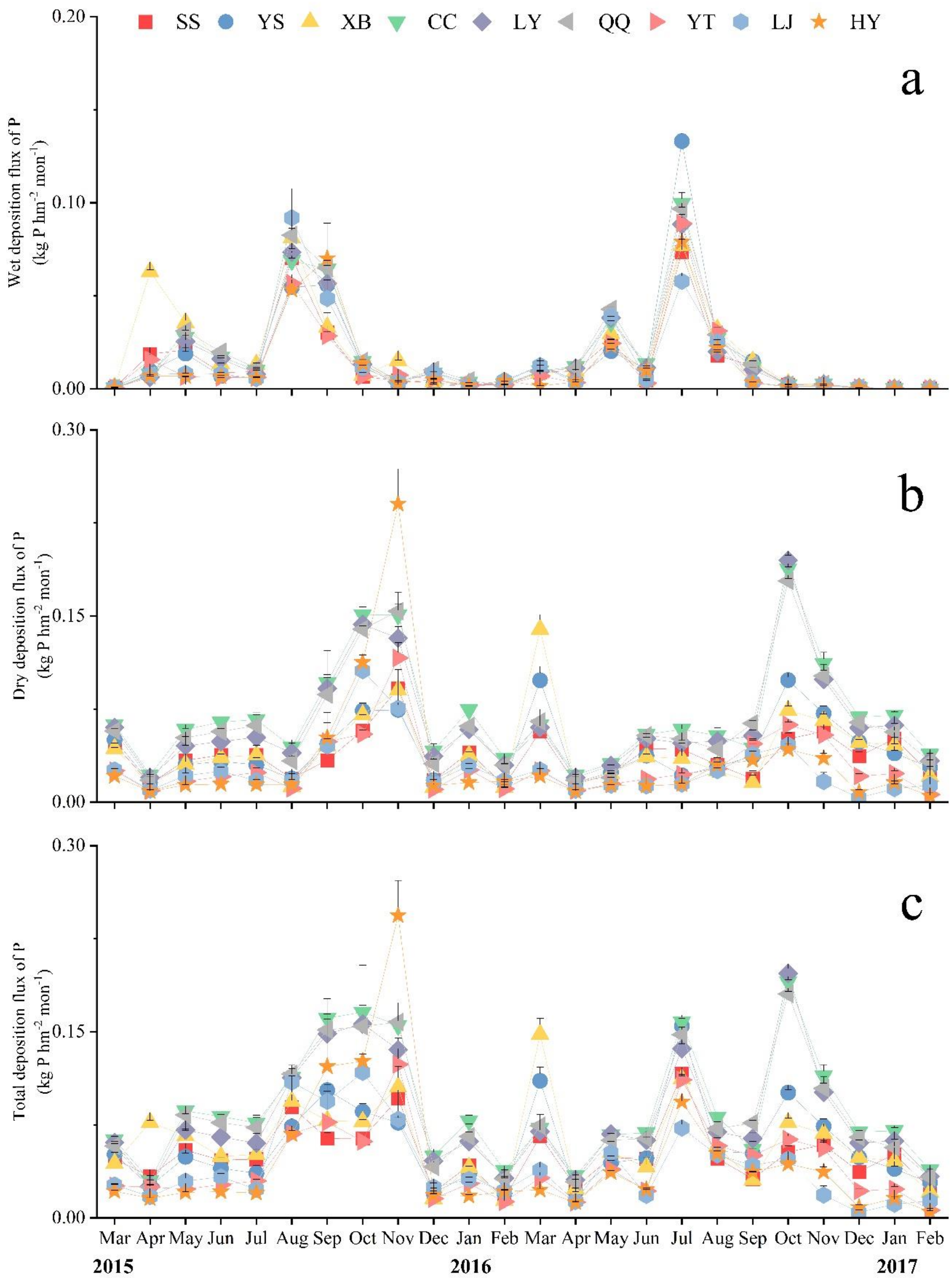

580 **Figure 2:** Monthly deposition fluxes of wet (a), dry (b) and total (c) deposition of P at nine study sites. Error bars represent the standard deviations of three replicates.

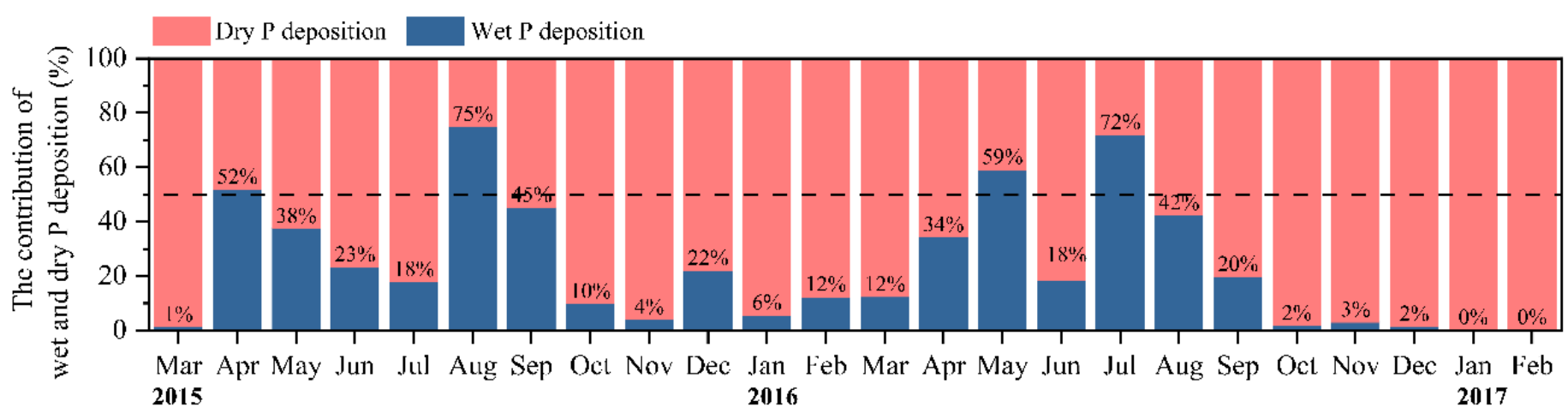

**Figure 3:** The contribution ratio of wet P deposition and dry P deposition to total P deposition. The middle-dashed line indicates each contributes 50%. The value represents the monthly contribution rate of wet P deposition to total P deposition.

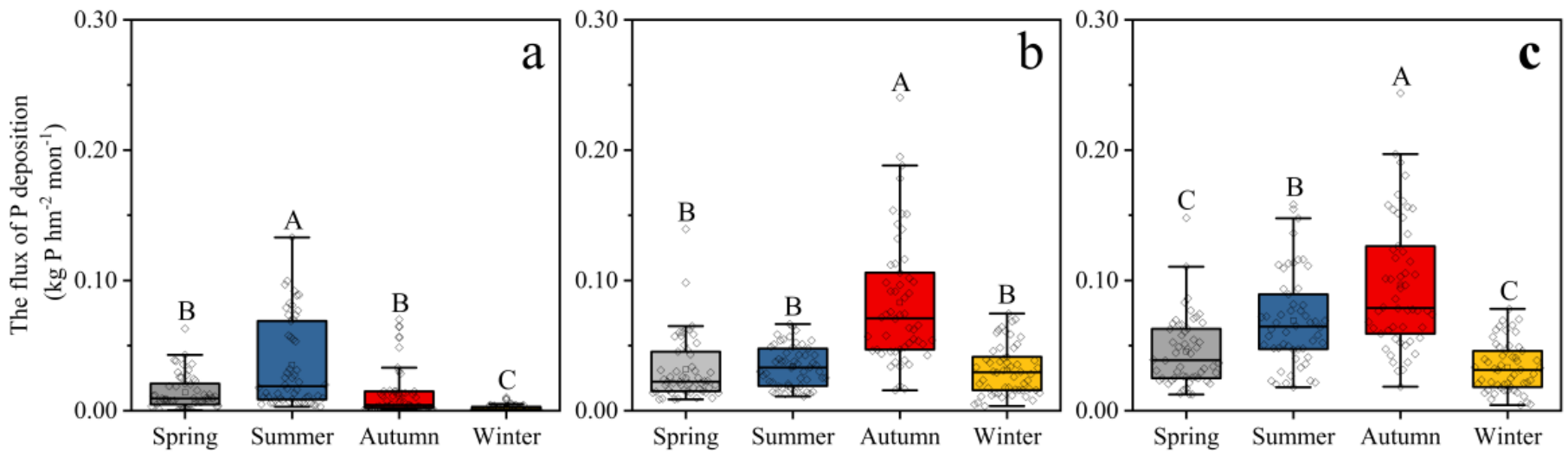

**Figure 4:** Monthly phosphorus flux of wet (a), dry (b) and total (c) deposition in four seasons. Different capital letters indicate that the differences among seasons are significant
590 (one-way ANOVA, $P < 0.05$).

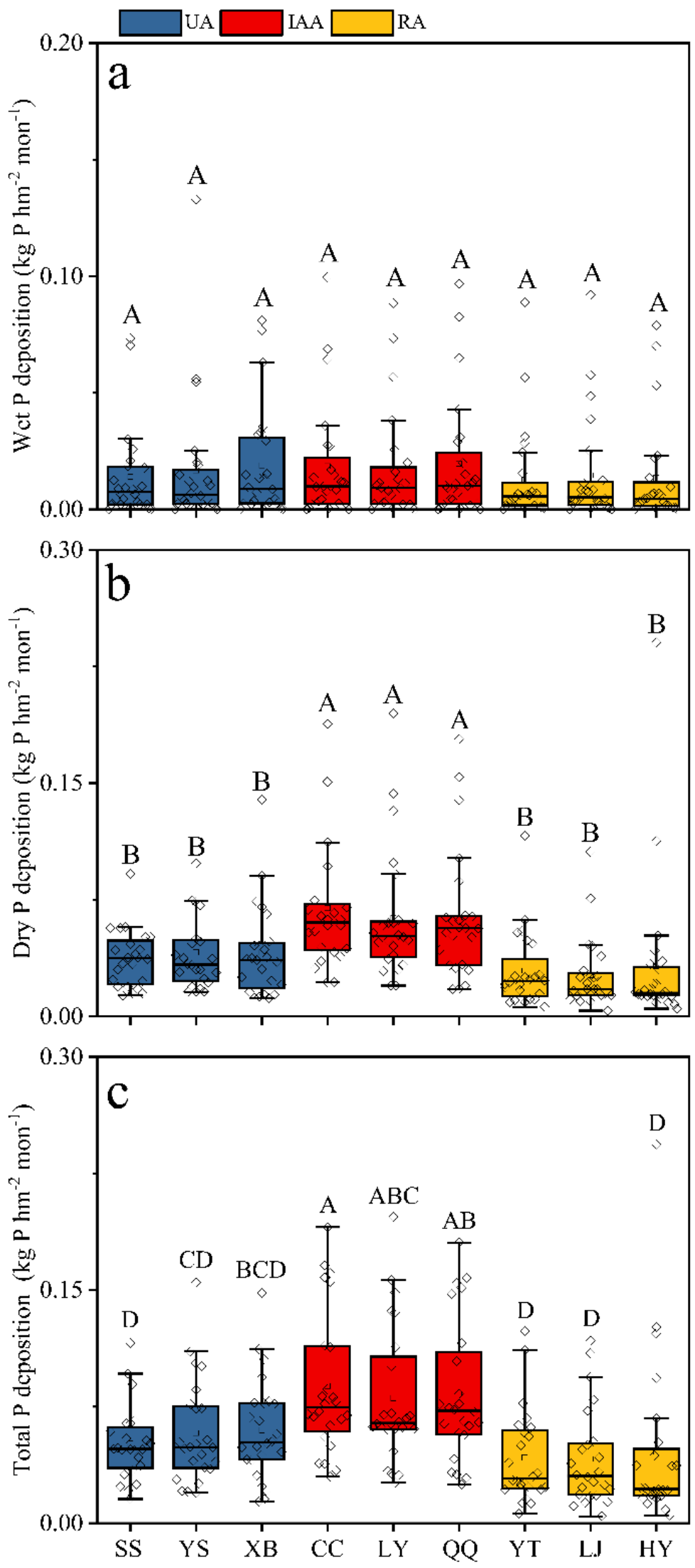

**Figure 5:** Average monthly phosphorus wet (a), dry (b) and total (c) deposition fluxes at nine sites. Each box contains 24 months of P deposition fluxes. Different capital letters suggest that the difference in the fluxes among the nine sites is significant ($p < 0.05$). (n=24 for each
595 box). In addition, different colored columns represent different areas.

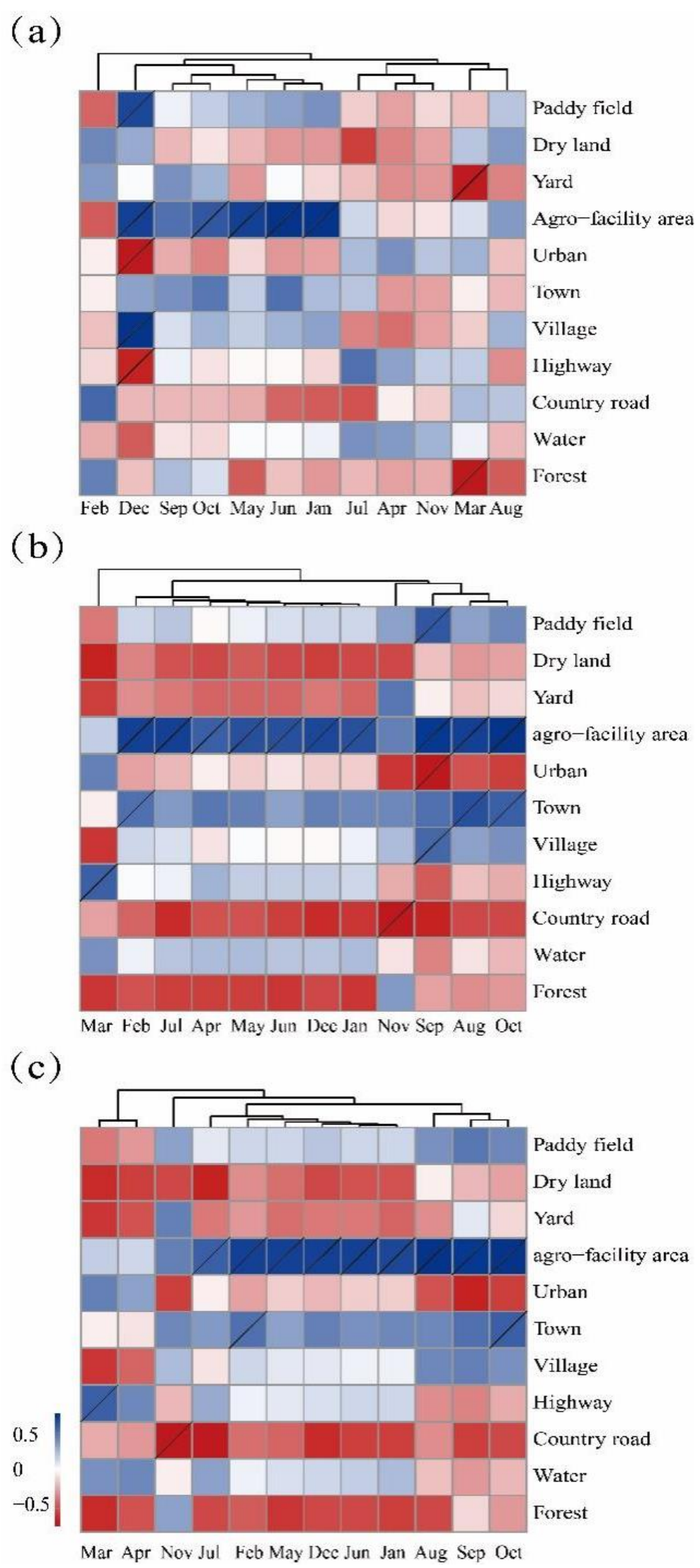

**Figure 6:** Pearson correlation between monthly wet (a), dry (b) and total (c) fluxes and areas of different land-use types. Gray slash indicates significance at $p < 0.05$.