# Peer review of "Fluxes, patterns and sources of phosphorus deposition in an urban-rural transition region in Southwest China"

_Atmospheric Chemistry and Physics, 2022_

## Author Response (AR1)

**Response to RC1's comments**

Dear Reviewer,

Thank you very much for helping us to handle the manuscript entitled "**Fluxes, patterns and sources of phosphorus deposition in an urban-rural transition region in Southwest China**" [**acp-2022-388**]. I am writing a response to the reviewer's comments. The detailed revisions are highlighted in yellow in the manuscript, and the response to comments are listed as follow:

**Q1: Source/sink (L311-315; L346-357; L388-389)**

A negative correlation between P deposition and country road/forest does not always indicate "sink" land use for P deposition. Because P deposition can be derived from soil particles containing P, a negative correlation indicates lower levels of P sources for P deposition in road and forest than in other land use such as agricultural areas. Although land-use of agriculture is intensive "source" for P deposition, the use of "sink" is carefully modified.

A: Thanks for your comments. Based on it, we make some corrections as follows:

The sentence "*Above all, the land use of "sink" denotes a lower level of P sources and a higher level of P sinks than other land use.*" was added on page 19, L365-366. We modified the sentence "*Based on correlation analysis, it was found that "source" land use might be agro-facility, town, and paddy field areas, while "sink" land use might be forest and country road areas.*" to "*Based on correlation analysis, it was 395 found that "source" land use might be agro-facility, town, and paddy field areas, while "sink" land use, which denotes a lower level of P sources and a higher level of P sinks than others, might be forest and country road areas.*" on page 20, L394-397.

**Minor comments**

**Q2: L13 phosphorus (P) à P**

A: We modified "phosphorus (P)" to "P" on page 2, L13.

**Q3: L19 76.13 à 76.1**

A: We modified "76.13" to "76.1" on page 2, L19.

**Q4: L45 I don't think that the term "in-situ" is needed in this manuscript. (also**

**L85, L378)**

A: We deleted "in-situ" on page 3-L45, page 5-L87, and page 20-L378.

**Q5: L51 What is "so caused"?**

A: The phrase "so caused" was modified to "the resulting" on page 4, L52.

**Q6: L73 "different" P-containing aerosols: What "different"?**

A: Thanks for your comments. What we would like to express here is the P-containing aerosols in different ways. We modified "different P-containing aerosols" to "P-containing aerosols originated from different ways" on page 5, L74-75.

**Q7: L102 Please clarify how to calculate the total area.**

A: Thanks for your comments. The sentence "*The total area of one land use type was calculated by adding the values in each column, as shown in Table 1, where each column indicates the area occupied by each land use type in nine sites.*" was added on page 6, L106-108.

**Q8: L125 Please provide how many samples for wet deposition were collected.**

A: We added the sentence "*During the sampling period, a total of 923 wet deposition samples were collected, including 858 valid samples.*" on page 7, L133-135.

**Q9: L144 P types: What "types"?**

A: We modified "P types" to TP on page 8, L149. We only measured the concentration of TP both in collecting wet and dry samples.

**Q10: L162 Please add the replication protocol in section 2.2 sample collection and analysis.**

A: Thanks for your comments. We added the sentence "*In addition, three parallel collectors were used at each site to collect atmospheric deposition to ensure three replicate data.*" on page 7, L116-117.

**Q11: L173 no data in February 2015 in Figure 2.**

A: Yes. This study was conducted from March 2015~February 2017, including 24 months.

**Q12: L179 25.0% to 99.7 % would be preferable.**

A: We modified it to "25.0% to 99.7 %". Please see on page 10, L184.

**Q13: L179 delete "generally"**

A: We deleted "generally" in L184.

**Q14: L184 Please define which months are categorized as summer.**

A: We have defined summer including June, July, and August in this study. Please see on page 10, L188-189.

**Q15: L191 I could not understand "three types of depositions".**

A: What we would like to express here is "atmospheric wet, dry, and total P deposition".

**Q16: L194 R=574: probably R=0.574**

A: We modified "R=574" to "R=0.574" on page 11, L199.

**Q17: L218 I could not understand "during almost the whole year".**

A: As shown in Fig.6, Pearson correlations between monthly dry (b) and total (c) fluxes and areas of the town and paddy field were positive during almost the whole year, except in March and April.

**Q18: L232 1.42-3.11 times higher: suggest "xx degree higher". Because temperature is also expressed as kelvin in addition to the degree.**

A: The expression "7.44-17.19 degrees higher" replaced "1.42-3.11 times higher" on page 13, L237.

**Q19: L233-235 The effect of temperature is not adequately discussed and supported.   Huang et al (2011) referred to in this study deal with sediment, not the atmosphere.**

A: We are sorry for our mistake. Through searching the references, we modified reference "(Huang et al., 2011)" to "(Tipping et al., 2014)" on page 13, L241.

**Q20: L237-249 Is there any possibility that P fertilizer was intensively emitted into the air in autumn, which enhances dry P deposition in autumn?**

A: We agree that P fertilizer was intensively emitted into the air in autumn. We discussed the driving effect of P fertilizer on dry deposition in section 4.4. Please check it on page 18, L349-355.

**Q21: L252 What deposition?**

A: "dry deposition" replaced "deposition" on page 14, L257.

**Q22: L263 What is "construction"?**

A: What we would like to express here is the contribution of wet and dry deposition to the total deposition.

**Q23: L298 Ling et al. (2022) is not listed in the reference list**

A: We are very sorry for our negligence of reference short citation. "Song et al. (2022)" replaced "Ling et al. (2022)" on page 16, L304.

**Q24: L298 I could not understand "this discrepancy". What differs, although P deposition is higher in agro-facility than in rural, urban, and forested areas, which seems similar to your study?**

A: Thanks very much for your comments. "This discrepancy" means this study's results of P deposition fluxes were higher than a large number of fluxes on a global scale (mentioned on page 15, L287-289). To accurately express "this discrepancy", we modified the sentence "*This discrepancy could be explained by the collection methods used for P deposition.*" to "*Furthermore, the collecting methods utilized for P deposition can also be used to explain the causes for the discrepancies between the experimental results of various studies.*" on page16, L305-306.

**Q25: L323 I could not understand "light".**

A: The word "lighting" replaced "light" on page 17, L332.

**Q26: L335 The correlation seems higher in August.**

A: The month "August" was added on page 18, L345.

**Q27: L358-375 I think the section of 4.5 Management practice of regional P showing surface water quality is not needed. While adequate data regarding P deposition is shown, the data on surface water quality is marginal.**

A: Thanks for your comment. We deleted it on page19, L359-368, and the sentence "*Since the P deposition in the study area is higher than in many regions, it should be monitored and controlled reasonably.*" was added on page 19, L374-375.

Dear Reviewer,

Thank you very much for helping us to handle the manuscript entitled "**Fluxes, patterns and sources of phosphorus deposition in an urban-rural transition region in Southwest China**" **[acp-2022-388]**. I am writing a response to the reviewer's comments. The detailed revisions are highlighted in yellow in the manuscript, and the response to comments are listed as follow:

**Specific comments**

**Q1: L20-21: This is incorrect. Correlation doesn't mean causality. Temperature and precipitation doesn't affect total P deposition! P emissions do!**

A: Thank you very much for your comments. We rewrite the sentence as:

"*Moreover, it was found that the monthly variations of P deposition were strongly correlated with meteorological factors, including precipitation, temperature, and relative humidity.*" on page2, L19-21.

**Q2: L24-25: It is well-known that dry P deposition is the primary form of total P deposition.**

A: Thank you very much for your comments. In this study, wet and dry deposition were measured separately. Therefore, we considered quantifying the ratio of dry and wet in this study area.

**Q3: L62-66: The correlations depend on whether total or wet deposition is analyzed. For instance, wet deposition correlates strongly with precipitation but total deposition doesn't.**

A: Based on your comments, we modified the sentence in L62-66 to "*Additionally, field studies have observed that the meteorological factors, including precipitation and temperature, could influence temporal variations of atmospheric P deposition (Tipping et al., 2014; Zhu et al., 2016; Chiwa et al., 2020).*" on page 4, L64-67.

**Q4: L79-80: Bulk deposition of P is likely very close to total P deposition.**

A: Thank you very much for your comments. Bulk deposition, which includes wet deposition plus a fraction of dry deposition. However, this study measured wet and dry separately and summed to obtain total P deposition. The method of collecting the

sample in this study is quite sparse, which is also mentioned in Tipping et al., 2014. Therefore, the difference between the two should be considered reasonable.

**Q5: L119-121 & 125: I don't believe you can do this manually for two years!**

A: Thank you for your affirmation! Since automatic monitoring is not possible at sampling sites, the manual collection is required. We informed manager site managers about collection methods and precautions to ensure accurate samples were collected.

**Q6: L157-158: Why a radius of 5 kilometers?**

A: The research group published a research article (Deng et al., 2019) which concluded: " N species deposition were significantly affected by the key land use types when radius were 3, 4 and 5 km". Based on this conclusion, we employ the largest radius (5km) of available land use data.

**Q7: L193-197 & Section 4.1: Correlation doesn't mean causality. See comments above.**

A: Thank you very much for your comments. We rewrite the sentence as: "*Moreover, it was found that the monthly variations of P deposition were strongly correlated with meteorological factors, including precipitation, temperature, and relative humidity.*" on page2, L19-21.

Besides, in section 4.1, we discussed how meteorological factors affect the process of P deposition.

**Q8: L215-216: Define agro-facility areas first.**

A: Thank you very much for your comments. We added the sentence "*Commonly, agro-facility areas include land designated for livestock and poultry breeding, fertilizer plants, greenhouses with vegetable production, and aquaculture (Current land use classification, GB/T 21010-2007).*" on page 4, L60-63.

**Q9: L258-261: Reference?**

A: Thanks for your suggestion. References "(Mahowald et al., 2008; Das et al., 2011; Gross et al., 2016)" was added on page 14, L265-266.

**Q10: L301: Bulk deposition is measured using a consistently open sampler. For P, I think bulk deposition includes wet deposition and a major proportion of dry deposition.**

A: Thank you very much for your comments. For most studies, P deposition was measured using a consistently open sampler, which collected bulk deposition. However, this study measured wet and dry separately and summed to obtain total P deposition. The method of collecting the sample in this study is quite sparse, which is also mentioned in Tipping et al., 2014. Therefore, the difference between the two should be considered reasonable.

**Q11: L308-309: Any details and evidences?**

A: Thank you very much for your comments. We modified the sentence "Moreover, the potential risk of P deposition in this study area cannot be ignored." to "*As discussed before, the flux of P deposition in this study area is at a high level. Excessive P deposition poses a certain threat to the ecosystem (Wang et al., 2015). Therefore, the potential risk of P deposition in this study area cannot be ignored.*" on page 16, L313-317.

**Q12: L350-351: Incorrect statement! Forest canopy strengthens deposition!**

A: We are very sorry for our mistake. We modified the statement to "*Notably, monthly fluxes of dry P deposition and total P deposition both had a negative correlation with forest and country roads (Fig. 6b, c). Firstly, a negative correlation indicates lower levels of P sources for P deposition in road and forest than in other land use such as agro-facility and agricultural areas. Secondly, it is well known that forests can absorb harmful gas, aerosols, and dust particles, including P-containing aerosols, which is attributed to the porous sponge-like underlying surface, high productivity, and strong microbial activity (Oladosu et al., 2017; Wang et al., 2017; Zhai et al., 2019). However, forest canopies could elevate P deposition by trapping atmospheric P in the form of dust and particulates (Zhou et al., 2018). Therefore, in this study, a negative correlation indicated that canopy P absorption was greater than trapping of atmospheric P (Parron et al., 2011). Above all, the land use of "sink" denotes a lower level of P sources and a higher level of P sinks than other land use. Due to similar reasons, paved country roads without hardening showed a similar correlation with P deposition.*" on page 18-19, L356-367.

**Q13: L370-374: References?**

A: We are very sorry for our negligence. Reference "Hochmuth et al., 2015" was added on page 20, L380, and "*Hochmuth, G., Rao, M., and Hanlon, A. E.: The four Rs of fertilizer management. UF/IFAS Extension, 1–4, https://edis.ifas.ufl.edu/publication/ss624, 2015.*" was added in reference lists, on page

24, L463-464.

**Finally, the ecological effects of P deposition are mediated by N deposition. What about N deposition in this region? This should be discussed somewhere.**

A: We agree with this viewpoint. The study of the N:P ratio is particularly important, especially in forest ecosystems. However, the purpose of this research is to understand the patterns of atmospheric P deposition in this region, from the perspective of temporal and spatial analysis. As a result, N deposition is not covered in this article. In subsequent investigations, we intend to further analyze and discuss N:P ratios in atmospheric deposition, water, and soil.

---

## Author Response (AR2)

**Dear Editors and Reviewers:**

Thank you for your letter and the reviewers' comments concerning our manuscript entitled "Fluxes, patterns and sources of phosphorus deposition in an urban-rural transition region in Southwest China" (acp-2022-388).

Those comments are all valuable and helpful for revising and improving our paper, as well as the important guiding significance to our research. Based on the comments, we made major revisions, please see the marked parts in the marked-up version. The main corrections in the paper and the responses to the reviewer's comments are as flowing:

**A response to the comments made by reviewer#1**

**Q1: That is why the sentence "the lowest fluxes were found in February 2015 (0.00~0.00 kg P hm$^{-2}$ mon$^{-1}$)" (L177-178 in the revised manuscript) should be deleted or modified.**

**A:** Thank you very much for your comments. We modified the sentence "*the lowest fluxes were found in February 2015 (0.00~0.00 kg P hm$^{-2}$ mon$^{-1}$)*" to "*the lowest fluxes were found in February 2017 (0.00~0.00 kg P hm$^{-2}$ mon$^{-1}$)*" on page 10, in L183.

**Q2: Adding "(atmospheric wet, dry, and total P deposition)" would be preferable after "three types of depositions", i. e., three types of depositions (atmospheric wet, dry, and total P deposition) (L195).**

A: Thank you very much for your comments. We added "*(atmospheric wet, dry, and total P deposition)*" on page 11, in L201-202.

**Q3: Suggest a modification like: "the contribution of wet and dry deposition to the total deposition was impacted by the solubility of P depositions and meteorological factors." (L268-269)**

A: Thank you very much for your comments. We modified the sentence "*the*

*contribution of depositions was impacted by the solubility of P depositions and meteorological factors.*" to "*the contribution of wet and dry deposition to the total deposition was impacted by the solubility of P depositions and meteorological factors.*" on page 14, in L275-277.

**A response to the comments made by reviewer#2**

**Q1:** The external input of anthropogenic phosphorus is important for natural ecosystems. To date, the measurements of dry phosphorus deposition are quite sparse than wet deposition or bulk deposition. While this manuscript aims to collect wet and dry deposition concurrently, which may enrich the database of the global phosphorus deposition pattern. However, the sampling method used in this study was not well designed and thus the results have large uncertainties. For example, the authors stated that "a cover on the top of the collectors was manually closed during rainfall events to eliminate influences from wet deposition", this seems unbelievable if done manually for nearly one thousand rain events during the two sampling years. In addition, the authors can not correctly sample wet deposition during rainfall events, if no automatic monitoring sampler was used, because dry deposited materials will collected with open sampler. As a result, the wet deposition in this study was not well separated from dry deposition, and vice versa, leading to large uncertainties in the observations. Overall, this manuscript is like a data report rather than research article, and the novelty is not well clarified yet.

**A:** Thanks very much for your comment. We know automatic monitoring samplers would produce more accurate data. But it is basically impossible for us to set up automatic samplers at 9 points due to financial constraints. The manual sampling method has been made to balance the need for accurate estimation and constraints in time and cost. In the old manuscript, the manual sampling method was not described clearly. We only collected dry deposition and wet deposition for 5 consecutive days per month, respectively, not for the whole month. And more than 1,000 mixed samples were collected continuously each month, with dry and wet deposition each accounting for half. Meanwhile, all sampler managers are trained and paid a monthly

management fee based on sample quality. And the results of this study showed that the wet deposition was close to the results of the study which collected deposition by an automated wet-dry sampler ( *He et al., 2011; He, J., Balasubramanian, R., Burger, D. F., Hicks, K., Kuylenstierna, J. C. I., and Palani, S.: Dry and wet atmospheric deposition of nitrogen and phosphorus in Singapore, Atmospheric Environment, 45, 2760–2768, https://doi.org/10.1016/j.atmosenv.2011.02.036, 2011.*), indicating that the manual sampling method did not overestimate wet deposition. More details of the sampling method were added in the new manuscript, please see it on pages 7-8, in Line 122 to Line 141.

*"Dry deposition was determined by the aqueous surface method (Anderson and Downing, 2006). Briefly, three pre-clean glass cylinders (inner diameter × height of 10.5 cm × 14.5 cm) were used as dry collectors at each site. All the collectors were placed 1.2 m above the ground with no obstacles and tall buildings around each site. A stainless-steel net (pore size, 0.02 × 0.02 $m^2$) was used to avoid any disturbance and pollution from birds and crops. The cylinder was filled with ultrapure water and examined if a refill was needed on 4 or 6 h basis (4 h in summer and 6 h in other seasons) to keep the water depth at a level of about 10 cm (Wang et al., 2016). Dry deposition sampling was conducted for five consecutive days at the end of the month, avoiding continuous rainfall as much as possible. Samples were collected in pre-clean glass bottles with lids at 8:00 am during these 5 days periods. In case of rainfall, the lid on top of the collector was manually closed to eliminate the effect of wet deposition. At the end of sampling every month, samples collected on 5 days were mixed and transported to the laboratory to determine total P (TP) concentrations on the same day.*

*Five consecutive days per month with a relative frequency of rainfall events were selected for wet deposition collection, based on weather forecasts every month. Wet deposition was collected at the end of each rainfall event (Oladosu et al., 2017), If the volume of samples (100 mL) collected in one rainfall event was too little, samples from continuous rainfall events were pooled as one mixed sample. The duration (min) and rainfall capacity (mm) were recorded for each rainfall event. Rainfall samples*

*collected monthly were mixed and transferred to the laboratory to determine total P (TP) concentrations on the same day.*

*During the sampling period, a total of 1026 deposition samples were collected, with half of the dry and half of the wet deposition samples. Changes in sample volume and air exposure were minimized. Moreover, river water samples from the Xihe River (103°39′57″ E, 30°36′02, XB) were collected to measure the P concentration.*"

In addition, supplementing the rather sparse database by direct monitoring of dry phosphorus deposition is one of the main objectives of this study, but a more important research objective of this study is to compare the differences in P deposition under multiple land uses and its causes. We believed that a basically unified sampling method can address the later questions. Few studies have analyzed the sources of P deposition based on the relationship with land use. This study found that the fluxes of dry P deposition were increased with the agro-facility, town, and paddy field areas, but decreased with the forest and country road areas, which was the novelty of this research article. It is important for understanding the process of regional P deposition and regional P management with "source/sink" land use.

Last but not least, your question is very reasonable. We agree with your comment that automated sampling will reduce the uncertainty of the results. Therefore, we have added automatic dry and wet deposition samplers at a sampling site (103°40′15″ E, 30°32′36″, QQ) where they are available, and have been collecting data on P deposition since February. However, historical data cannot be re-collected, and the automatic collector cannot cover the entire urban-rural transition zone, so the data from this study are still important and valuable.

---

## Author Response (AR3)

**Dear Editors:**

Thank you for your comments concerning our manuscript entitled "Fluxes, patterns and sources of phosphorus deposition in an urban-rural transition region in Southwest China" (acp-2022-388).

Those comments are all valuable and helpful for revising and improving our paper, as well as the important guiding significance to our research. Based on the comments, we made minor revisions, please see the marked parts in the marked-up version. The main corrections in the paper are as flowing:

**Q1: In Eq. (1) (lines 145-150), you used the number of non-rainy days per month for N to calculate the total dry deposition. Please note that in reality dry deposition happens all the time, even during precipitation. And If fact, dry deposition velocity is larger (for most chemical species) on rainy than dry days due to the stronger turbulence on rain days. Even though in the field measurement of dry deposition rainy days are typically excluded, in theoretical calculation rainy days should be included in dry deposition calculation. Please make necessary changes and revise the deposition numbers accordingly.**

**A:** Thank you very much for finding this mistake in our manuscript. In general, the total days per month were used to calculate total dry depositions. We checked our original data and ensured that the total number of days per month for N was used, but not "non-rainy" days per month. There is a certification that July 2015, which was the least "non-rainy" month (around 7 days), showed high values of dry deposition. We corrected this mistake in the new manuscript, on page 8, in Line 149 as follows. Thanks again for pointing out this mistake.

"*N is the total number of days per month, d;*"

**Q2: Line 394-395 (and the paragraph starting on line 147): In theory dry deposition is mainly controlled by ambient concentration and dry deposition velocity, with the latter being more affected by meteorological factors. RH may**

**play a factor in this, but is it the dominant factor? You should first state if seasonal variations in ambient concentration is large or not. You should first identify dominating factors, rather than just stating one parameter is related to this process.**

A: Thank you very much for your comments. We agreed that RH indirectly affects dry deposition by increasing the particle deposition rate. And we cannot sure that RH is the dominant factor of dry deposition, given that dry deposition also depends on ambient concentration. Therefore, we added these sentences on page 13, in Lines 248-253 as follows:

"*In theory, the seasonal variation of dry deposition is mainly controlled by ambient concentration and dry deposition velocity. On one hand, high ambient concentrations could be caused by high ambient emissions. In this study, the application of P fertilizer was conducted in Autumn, causing higher ambient emissions than in other seasons. Another hand, dry deposition velocity was more affected by meteorological factors.*"

In addition, we rewrote the last sentence of this section and the related conclusion, as follows:

"*In contrast, dry deposition was influenced by relative humidity and ambient concentration.*" on page 14, in Lines 263-264.

"*While dry deposition was affected by relative humidity and ambient concentration and was significantly higher in autumn (P < 0.05).*" on page 20, in Lines 399-401.